# Bacterial Diversity and Potential Functions in Response to Long-Term Nitrogen Fertilizer on the Semiarid Loess Plateau

**DOI:** 10.3390/microorganisms10081579

**Published:** 2022-08-05

**Authors:** Aixia Xu, Lingling Li, Junhong Xie, Renzhi Zhang, Zhuzhu Luo, Liqun Cai, Chang Liu, Linlin Wang, Sumera Anwar, Yuji Jiang

**Affiliations:** 1State Key Laboratory of Aridland Crop Science, Gansu Agricultural University, Lanzhou 730070, China; 2College of Agronomy, Gansu Agricultural University, Lanzhou 730070, China; 3College of Resource and Environment, Gansu Agricultural University, Lanzhou 730070, China; 4Institute of Molecular Biology and Biotechnology, The University of Lahore, Lahore 54000, Pakistan; 5State Key Laboratory of Soil and Sustainable Agriculture, Institute of Soil Science, Chinese Academy of Sciences, Nanjing 210008, China

**Keywords:** N fertilizer, microbial community, soil bacteria, high-throughput sequencing, PICRUSt

## Abstract

Bacterial diversity and its functions are essential to soil health. N fertilization changes bacterial communities and interferes with the soil biogeochemical N cycle. In this study, bacterial community and soil physicochemical properties were studied in 2018 after applying N fertilizers (0, 52.5, 105, 157.5, and 210 kg N ha^−1^) for a long (2003–2018) and a short (2003–2004) duration in a wheat field on the Loess Plateau of China. Soil bacteria were determined using 16S rRNA Illumina-MiSeq^®^, and the prediction function was analyzed through PICRUSt. The study showed that N fertilizer significantly changed the diversity and abundance of bacterial communities. The phyla *Proteobacteria*, *Actinobacteria*, *Acidobacteria*, and *Chloroflexi* were most abundant, accounting for 74–80% of the bacterial community abundance. The optimum rates of N fertilizer application (N105) maintain soil health by promoting soil microbial diversity and abundance. The bacterial population abundance was higher after short-term N application than after N application for a long duration and lowest with the high N-fertilizer treatment (N210). High N enrichment led to more heterotrophic N-fixing microorganisms (*Alphaproteobacteria*), in which metabolism and genetic information processing dominated, while cellular processes, genetic information processing, metabolism, and organismal systems were the main functional categories under low N. The phyla *Gemmatimonadetes*, *Actinobacteria*, *Bacteroidetes*, and *Chloroflexi* were the key bacteria in the co-occurrence network. The genus *Saccharimonadales* of the superphylum *Patescibacteria* has a more significant impact under low N treatment. Long-term N fertilization affected the soil pH, NO_3_-N, and other physicochemical properties, and soil NO_3_-N was the highest indicator, contributing 81% of the bacterial community function under different N fertilizer treatments.

## 1. Introduction

Microorganisms are the most active part of the soil ecosystem as they transform and recycle soil nutrients [1] by the decomposition of soil organic matter [2]. Microbial community diversity and composition are indicators of soil fertility and health [3]. Microbial diversity is crucial for maintaining soil quality [4]; in particular, high bacterial diversity is essential for sustainable agriculture [5] because of their role in nutrient cycling, disease suppression, and plant growth [6].

The bacterial community is particularly sensitive to changes in the soil environment [7]. Therefore, the diversity and structure of bacterial communities can be used to assess soil quality [8] and changes in soil nutrients [9]. Researchers found that *Proteobacteria*, *Actinobacteria*, and *Acidobacteria* are the main bacterial phyla [10,11], and *Agrobacterium*, *Alcaligenes*, and *Arthrobacter* are the most important bacterial genera in the soil [12].

Extensive agricultural production with high inputs of chemical fertilizers has threatened soil quality [13]. Nitrogen (N) is the main component of nucleic acid and protein and also a key nutrient for all biological communities [14]. Therefore, N is the most extensively used fertilizer for farmland ecosystems; its input is expected to be 112 million tons in 2022 [15,16].

N fertilization can effectively increase soil N content, thereby promoting the growth and reproduction of soil microorganisms [17]. However, long-term N fertilization and high N levels could change microbial community structure and activity, which in turn affect N status in the soil and loss of microbial diversity [18]. Zhou et al. [19] showed that long-term N fertilizer application had turned a healthy bacterial-type soil to a fungal-type soil that was prone to soil-borne diseases. Soil N content also affects nitrogen-oxidizing, nitrogen-fixing, and denitrifying bacteria and other N cycle functional flora, as well as those in carbon and phosphorus cycles [20,21].

Researchers showed that long-term N addition reduced soil pH [22,23]. Soil microorganisms are sensitive to pH changes [24], so N fertilizer may modulate change in microbial activity by affecting soil pH [25]. Long-term N fertilization changes soil physicochemical properties [21] increases residual nitrate in the soil, which can cause N leaching or loss, thereby causing groundwater pollution [26] or increasing greenhouse gas emissions [27]. However, how long-term N fertilization (especially residual N in soil) affects the soil bacterial community diversity and function is poorly understood.

The demand for sustainable land use in agricultural systems raises the need to understand plant–soil–microbe interactions and how to improve soil ecosystems [28]. Long-term fertilization experiments play a pivotal role in providing bases for decision-making related to sustainable agriculture because of its rich, accurate, and reliable data and strong explanatory power [29]. Due to the complexities of soils, the variety of microorganisms, and the limited cultivable microorganisms, long-term localization experiments on microorganisms, especially in crop fields, are still limited.

In this study, changes in bacterial community diversity were examined in wheat fields of the Loess Plateau where different rates of N fertilizers were applied for a continuous 16 years (long-term) or for 2 years (short-term). Bacterial diversity was determined using high-throughput 16S rRNA Illumina-MiSeq^®^ (San Diego, CA, USA), and the prediction function was analyzed through PICRUSt analysis. The study was carried out based on the hypothesis that N fertilization could influence the soil microbial community structure, diversity, and potential functions by changing the soil physicochemical properties. The specific goals were (1) determine that continuous N fertilization over 10 years may affect the soil bacterial community, (2) characterize the bacterial community and prediction function related to N fertilization, and (3) confirm that N fertilization will change soil microbes by changing the soil physicochemical properties rather than just soil pH.

## 2. Materials and Methods

### 2.1. Site, Experimental Design

A mono-cropping spring wheat experiment field was built in Dingxi, Gansu Province, China (35°28′ N, 104°44′ E, elevation 1971 m a.s.l.). The average daily minimum temperature of the experimental field was −22 °C in January, and the maximum temperature was 38 °C in July during 2003 to 2018. The annual average precipitation, evaporation, and radiations were 390.7 mm, 1531 mm, and 5930 MJ m^−2^, respectively. The field experimental soil was Calcaric Cambisoll [30], also known as Huangmian [31]. The soil is sandy loam, slightly alkaline (8.33 pH), with total N of 0.77 g kg^−1^.

A randomized block design was used to arrange the treatments, and the plot area was 3 m × 10 m, repeated 3 times. Five treatments, i.e., non-N-fertilized control (N0), 52.5 (N52.5), 105 (N105), 157.5 (N157.5), and 210 (N210) kg N ha^−1^ were applied in the form of urea. In the first main plot, N fertilizer rates were applied from 2003 to 2018. In the second main plot, the N treatments (N52.5c, N105c, N157.5c, and N210c) were applied for two years in 2003 and 2004. Enough K content (221.42 mg kg^−1^ available K) was present in the soil (60 cm layer) of the experimental field, so no potassium fertilizer was applied. Each treatment applied the same amount of calcium superphosphate (105 kg P_2_O_5_ ha^−1^), and all fertilizers were evenly spread on the entire subplot according to the treatment requirements at sowing. In all studied years, spring wheat (Dingxi 38, 187.5 kg ha^−1^ seeding rate, 20 cm row spacing) was planted in mid-March and harvested around late July of the same year.

### 2.2. Samples, Physicochemical and Bioinformatic Analysis

In 2018, soil samples were collected at the anthesis and grain filling period (14 DAA) of wheat at a depth of 0–20 cm in a zig-zag pattern and sieved (2 mm) after mixing. A part of the mixed soil was air-dried for physicochemical analysis, and the other was shipped using a sealed box with dry ice for molecular analyses.

Soil pH was detected by glass electrode method (soil: water = 1:2.5) [32]. Total N was detected by Kjeldahl method. Available phosphorus was detected by colorimetry [32]. NH_4_-N and NO_3_-N were determined by spectrophotometry [33]. Moisture was determined gravimetrically [32].

Then, bacterial DNA was extracted using DNeasy PowerSoil Kit (No. 12888-100, Qiagen, North Rhine-Westphalia, Hilden, Germany) per instructions. The V3/V4 heterogenous gene region used primers 343F-5′-TACGGRAGGCAGCAG-3′ and 798R-5′-AGGGTATCTAATCCT-3′ to analyze bacterial diversity. Illumina MiSeq sequencing was used to generate double-ended sequence raw data. The split_libraries (version 1.8.0) [34] software in QIIME was used to obtain high-quality clean tags, UCHIME (version 2.4.2) [35] to obtain valid tags, and Vsearch (version 2.4.2) [36] to cluster valid tags into operational taxonomic units (OTU) (97% similarity); these were used for subsequent analysis.

### 2.3. Data Analysis

SPSS 23.0 software (Chicago, IL, USA) was used to analyze the differences between treatments (*p* < 0.05). Alpha diversity analysis (Shannon, Chao1, and PD_whole_tree diversity indexes) was used to assess community richness and homogeneity. Beta diversity analysis was used to analyze community compositions evaluated by principal coordinate analysis (PCoA). PICRUSt software was used to predict microbial gene functions [37]. The Kyoto Encyclopedia of Genes and Genomes (KEGG) was used to assign and compare functional counts and annotations [38]. Spearman’s correlation and the Kullback–Leibler dissimilarity (KLD) measure was used to construct co-occurrence networks to describe the symbiosis mode in the soil bacterial network [39]. The co-occurrence network was conditioned by the Conet plug-in in Cytoscape software and visualized in Gephi software [40]. Multiple relationships between soil physicochemical properties and distribution of dominant phyla were evaluated by distance-based redundancy analysis (RDA). Indicator analysis was used to calculate the OTU in each sample, and then the indicator value was statistically analyzed (*p* < 0.05) to reveal the biological species or genera.

## 3. Results

### 3.1. Soil Bacterial Alpha Diversity

Illumina MiSeq analysis showed that 1,124,670 filtered sequences (ranging from 33,244 to 49,879 reads per sample) and 901,159 reads were generated and clustered into 4613 OTUs, of which 353 OTUs, 25 genera, and 6 phyla were significantly different (Appendix A). The Good’s coverage index (>0.967) showed a reasonable amount of sequencing data and sufficient sequencing coverage. Therefore, OTUs were representing the entire microbial community library.

Alpha diversity indices indicated significant differences in bacterial populations among the treatments (Figure 1). The Chao1, PD_whole_tree, and OTU richness indexes were highest for the soil treated with N105, indicating that the N105 treatment resulted in higher bacterial population abundance. The Shannon index was lowest for N210 soil samples, suggesting that the higher N fertilizer lessened the bacterial population abundance. The Chao1 and OTU richness indexes of N52.5c, N105c, N157.5c, and N210c had no significant differences compared with N0, indicating that the bacterial population was not affected by short-term N and not supplying N after a short-term application.

It was also observed that bacterial OTUs were most abundant in N105 and least in N210, and the order was N105 > N157.5 > N157.5c > N105c > N52.5 > N52.5c > N210c > N0 > N210 (Appendix A).

### 3.2. Soil Bacterial Beta Diversity

The principal coordinate analysis (PCoA) showed that total variances by PCo1, PCo2, and PCo3 based on the weighted UniFrac were 39.7%, 20.3%, and 9.9% (Appendix A), and those based on the unweighted UniFrac were 9.7%, 5.8%, and 5.3%, respectively (Appendix A).

The application of N fertilizer for 16 y significantly impacted the relative abundance of *Proteobacteria*, *Actinobacteria*, *Chloroflexi*, *Gemmatimonadetes*, *Nitrospirae*, *Verrucomicrobia*, and *Planctomycetes* (Figure 2, Appendix A). The dominant phyla in the soil bacterial community were *Proteobacteria* (23–29%), *Actinobacteria* (19–21%), *Acidobacteria* (17–27%), and *Chloroflexi* (11–14%), which accounted for 74–80% relative abundance, followed by *Gemmatimonadetes*, *Bacteroidetes*, *Nitrospirae*, *Verrucomicrobia*, and *Planctomycetes*, with lower abundance. Soil bacterial diversity at N105 and N210c was similar. The relative abundance of *Verrucomicrobia* in the higher-N treatments was 0.4–0.57 times and 0.92–1.15 times lower at N157.5 and N210, while *Proteobacteria* at N210 increased 0.07–0.27 times more than other treatments; *Gemmatimonadetes* in the lower-N treatment (N52.5c) and no-N-fertilized control (N0) was less (0.08–0.19 times and 0.09–0.20 times) than that of other treatments; *Acidobacteria* and *Planctomycetes* were higher in N105, N210c, and N0 than other treatments; *Nitrospirae* was increased (0.09–0.32 times) by N105 than by other treatments. Furthermore, the relative abundance of class *Spartobacteria* and *Phycisphaerae* in the N105 increased 0.081–1.44 times and 0.01–0.87 times and in N210c increased 0.07–1.40 times and 0.07–0.99 times, both respectively (Appendix A).

### 3.3. Function Prediction

PICRUSt explored the first (Table 1) and second levels (Table 2) of metabolic functions of soil bacterial communities. PICRUSt analysis identified seven functions in the first level of Kyoto Encyclopedia of Genes and Genomes (KEGG) (Table 1). The first level of KEGG predictive function indicated that the metabolism and genetic information processing were the dominant functions, accounting for 42% and 18% of the total abundance of functional genes, respectively. Human disease was maximum under the highest-N treatment (N210). N105 treatment had the highest level of cellular processes, genetic information processing, metabolism, and organismal systems. The heatmap clustering showed the close segregation of N210 with N210c and of N157.5 with N105 (Figure 3).

At the second level of the functional spectrum of soil bacterial communities, 8 groups of 41 KEGG pathways were significantly different (Table 2). The highest N treatment (N210) showed significantly high enrichment in cell growth and death, signaling molecules and interaction, xenobiotics biodegradation and metabolism, circulatory system, and cellular processes and signaling and less glycan biosynthesis and metabolism and environmental adaptation. The short-term N application (N105c, N157.5c, and N210c) showed more signaling molecules and interaction and genetic information processing compared with the non-N control. The N210 treatment increased cell growth and death, signaling molecules and interaction, xenobiotics biodegradation and metabolism, circulatory system, and cellular processes and signaling by 7.4%, 16.9%, 4.4%, 15.5%, and 3.1%, respectively, compared with the non-N control. The N105 treatment increased genetic information processing compared with N0 and N210 treatment. N105c, N157.5c, and N210c treatments increased signaling molecules and interaction and genetic information processing compared with N0.

### 3.4. Bacterial Co-Occurrence Networks and Prediction Analysis

In the bacterial network, there were 40 nodes assigned to classes *Gemmatimonadetes* (9.1%), *Actinobacteria* (7.3%), *Bacteroidetes* (1.8%), and *Chloroflexi* (1.8%) (Figure 4). In addition, there were 25 edges in the bacterial network, and the average path length was 1.4.

Random forest importance of species (variables) points showed that the genera *Blastocatella*, *Gaiella*, *Chthoniobacter*, and *Flavisolibacter* had the most importance of species (variables) points (>1) across soil samples (Figure 5). Indicator analysis showed that genus *Saccharimonadales* of phylum *Patescibacteria* has a greater impact on the growth environment with N105 treatment (Appendix A).

### 3.5. Physicochemical Properties and Bacterial Community

Soil indictors (pH, NH_4_-N, NO_3_-N, and AP) differed significantly among treatments (Table 3). The soil pH decreased (8.09 to 7.81) with different N application rates, and pH at all N-fertilizer treatments was significantly lower than N0. N157.5 and N210 had significantly lower pH than the lower-N treatments (N52.5). Soil NO_3_-N and available phosphorus (AP) increased by 15–67% and 9–19%, respectively, with N treatment. Both the continuous long-term and short-term application of N increased the soil NO_3_–N, but the increase was greater after long-term application. The soil NH_4_–N increased with N application over the control, but significant increases were only observed at N157.5 and N210. Soil available phosphorus (AP) increased with N application over control, and the maximum AP was observed sat N105.

RDA analysis showed that 53.5% of the total variation was explained by axis 1 and 4.3% by axis 2 (Figure 6). The phyla *Saccharibacteria*, *Proteobacteria*, *Bacteroidetes*, *Gemmatimonadetes*, and *Nitrospirae* were clustered together and positively influenced by the soil NO_3_–N and NH_4_–N as well as negatively related with soil pH. Among the soil physiochemical properties, NO_3_–N explained 49.8% of the variation and was the most significant predictor with 80.7% contribution (Appendix A). The phyla *Verrucomicrobia*, *Acidobacteria* and *Chlorflexi* was negatively related with NO_3_–N and NH_4_–N.

Low soil NO_3_-N and NH_4_-N decreased the phyla Proteobacteria and Bacteroidetes, and increased phyla *Acidobacteria*, *Chloroflexi*, *Verrucomicrobia*, and *Planctomycetes* (Table 4). Low soil pH decreased phyla *Actinobacteria*, *Acidobacteria*, *Chloroflexi*, and *Verrucomicrobia* and increased phyla *Proteobacteria*, *Gemmatimonadetes*, *Bacteroidetes*, and *Nitrospirae*.

## 4. Discussion

A soil’s bacterial community is a health indicator usually affected by multiple factors such as fertilization, agricultural practices, and environmental conditions [6,10]. Previous studies have shown that short-term application of N fertilization either leaves bacterial diversity unchanged or only shows temporary changes [23,41]. However, long-term N fertilization resulted in significant changes in bacterial community structure and diversity, resulting in a decline in bacterial diversity [42,43]. The present results show that the application of different rates of N fertilizer for 16 years significantly affected the soil physicochemical properties and the composition of the soil microbial community. However, the microbial diversity was lost as a consequence of long-term excessive N fertilizer application [43,44]. We found that the optimum rates of N fertilizer application maintain soil health by promoting soil microbial diversity and abundance, as bacterial population abundance was higher at the optimum N-fertilizer (N105) [45] and lowest at the high N-fertilizer treatment (N210). The bacterial population abundance was higher after short-term N application than after N application for a long duration. After two years of application of different N fertilizer rates and then not applying N fertilizer for several years (14 years) (N52.5c–N210c), the bacterial population abundance tends to be the same as in the non-fertilized control (N0).

In this study, the alpha diversity of soil bacteria was greatest with N105 treatment: N105 treatment resulted in greater bacterial population abundance, while the alpha diversity index at N210 treatment was lower than that of non-N control, indicating that higher N-fertilizer rate led a decrease in bacterial population abundance (Figure 1). The analysis of the taxonomic components of microorganisms showed that the phylum *Proteobacteria* (24.8%) was the most predominant bacterial, followed by *Actinobacteria* (20.1%), *Acidobacteria* (19.6%), and *Chloroflexi* (13%) (Figure 2, Appendix A), and the similar trend was also observed by some previous studies [44,46].

N enrichment in soil affects the composition and function of soil microbial communities [47,48] such as increasing trophic taxa and reducing oligotrophic taxa of soil [49]. High N could even reduce microbial diversity and abundance [50,51]. The phylum *Proteobacteria* is considered as a copiotrophs [52], and therefore higher N treatment (N210) showed a high relative abundance of *Proteobacteria*. In contrast, higher N treatment (N210) lowered the relative abundance of *Acidobacteria*, because *Acidobacteria* are oligotrophs [53]. The bacteria of class *Alphaproteobacteria* are heterotrophic nitrogen-fixing microorganisms [54], and their relative abundance was highest with the highest N treatments (N210) due to higher residual soil TN [54]. In addition, there were similar soil physicochemical properties and bacterial community structure under N105 and N210c treatments with a relative higher abundance of phyla *Acidobacteria*, *Planctomycetes*, and *Nitrospirae*, suggesting that N had a significant residual effect on bacterial communities.

The PICRUSt function prediction analysis was used to determine the metabolic and functional capability [55,56]. Metabolism and genetic information processing were the dominant groups of functions, as revealed by PICRUSt analysis, accounting for 42% and 18% of total abundance of functional genes, respectively. The highest human disease function occurred with the highest N treatment (N210), while the most cellular processes, genetic information processing, metabolism, and organismal systems occurred at the medium N rate (N105), reflecting that the optimum-N fertilizer application makes soil healthier than the higher N rate. Functional genes indicate the relative abundance of certain types of microorganisms that are influenced by the biogeochemical cycles of major elements, i.e., C, N, and P [57]. Furthermore, the cellular processes of cell growth and death, environmental information processing of signaling molecules and interaction, metabolism of xenobiotics biodegradation and metabolism, and organismal systems of circulatory system were significantly greater, while glycan biosynthesis and metabolism and environmental adaptation were significantly lower with the highest N treatment (N210) compared with the non-N control.

The interactions among microorganisms such as mutual benefits or competition influence the nutrient cycle and soil health [58]. The symbiosis network between microorganisms reflects the function and stability of a biological community [59,60]. Co-occurrence network analysis analyzes the key interactions between microorganisms and their responses to the environment to explore the basic principles of the microbial communities of different soil environments and identify key groups assumed in the communities [61,62]. *Proteobacteria* and *Actinobacteria* are the main representative bacterial communities that have roles in biological carbon and nitrogen cycles [57]. This study observed that *Gemmatimonadetes*, *Actinobacteria*, *Bacteroidetes*, and *Chloroflexi* are the main components in the co-occurrence network.

The changes in microbial communities are mainly driven by soil characteristics such as soil pH, organic carbon, and N [63,64]. Fertilizer addition changes the microbial community composition and structure by effecting various soil physicochemical properties [47,65]. Soil microorganisms perceive changes in soil properties caused by N fertilizer and respond differently [23,66]. We found that the soil pH, NH_4_–N, NO_3_–N, and AP had significant differences between different N fertilizer treatments. N fertilization resulted in a decrease in soil pH from 8.09 to 7.81. In addition, after applying different rates of N fertilizer for two consecutive years, soil pH was still significantly lower than that of the non-N control after no application of N fertilizer for 14 years. Studies have shown that soil bacteria are most affected by pH [66]. Therefore, soil acidification by N addition is used to explain the observed decline in soil bacterial diversity [46,50,67].

The present study showed that the low soil pH decreased the abundance of phylum *Actinobacteria* and increased the abundance of *Nitrospirae*. These results were similar to previous findings indicating that the low soil pH decreased the abundance of *Actinobacteria*, *Proteobacteria*, and *Bacteroides* caused by soil acidification [68] and increased the abundance of *Acidobacteria* and *Nitrospirae* [69,70]. However, in contrast to previous results, the low soil pH decreased the abundance of *Acidobacteria* and increased the abundance of phyla *Proteobacteria* and *Bacteroidetes*, which might be due to the weakly alkaline soils in this field (pH ranged from 7.81 to 8.09). Since bacteria can tolerate a narrow pH range [68], they are more sensitive to pH. However, it has not been definitively confirmed whether the soil pH has a direct or an indirect effect on the bacterial community. soil NO_3_–N is one of the highly responsive indicators of the bacterial community function under different N fertilizer treatments (contribution of 80.7%) [23,66]. Bacterial communities are affected by multiple soil physicochemical properties such as NO_3_–N, NH_4_–N, and soil moisture.

## 5. Conclusions

N fertilization for 16 years significantly affected the soil NO_3_–N, thus changing the diversity and abundance of bacterial communities. The phyla *Proteobacteria*, *Actinobacteria*, *Acidobacteria*, and *Chloroflexi* accounted for 74–80.3% of the bacterial community abundance. N fertilization at high rates reduced bacterial community diversity and led to more heterotrophic N-fixing microorganisms (*Alphaproteobacteria*), in which metabolism and genetic information processing dominated. The relative abundance of *Proteobacteria* was increased while *Acidobacteria* was reduced, and the abundance of heterotrophic N-fixing microorganisms increased due to greater residual soil TN. On the other hand, the optimum rate of N fertilizer enhanced the soil health by promoting soil microbial diversity and abundance, and the genus *Saccharimonadales* of phylum *Patescibacteria* was the biological genus that had the most impact on its growth environment.

## Figures and Tables

**Figure 1 microorganisms-10-01579-f001:**
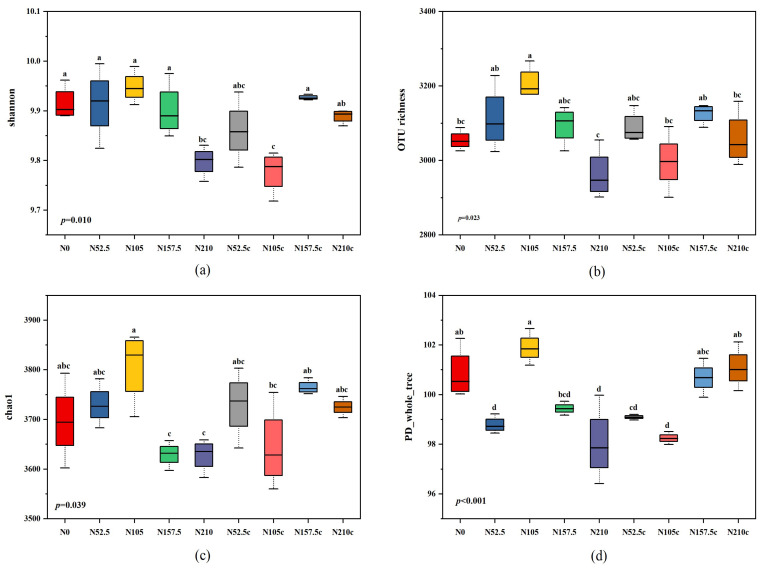
Alpha diversity analyses of Shannon (**a**), OTU richness (**b**), Chao1 (**c**), and PD_whole_tree (**d**) indexes. N0, 0 kg N ha^−1^ year^−1^; N52.5, 52.5 kg N ha^−1^ year^−1^; N105, 105 kg N ha^−1^ year^−1^; N157.5, 157.5 kg N ha^−1^ year^−1^; N210, 210 kg N ha^−1^ year^−1^; N52.5c, 52.5 kg N ha^−1^ 2 year; N105c, 105 kg N ha^−1^ 2 year; N157.5c, 157.5 kg N ha^−1^ 2 year; N210c, kg N ha^−1^ 2 year. Different letters above the boxes are significant at *p* < 0.05.

**Figure 2 microorganisms-10-01579-f002:**
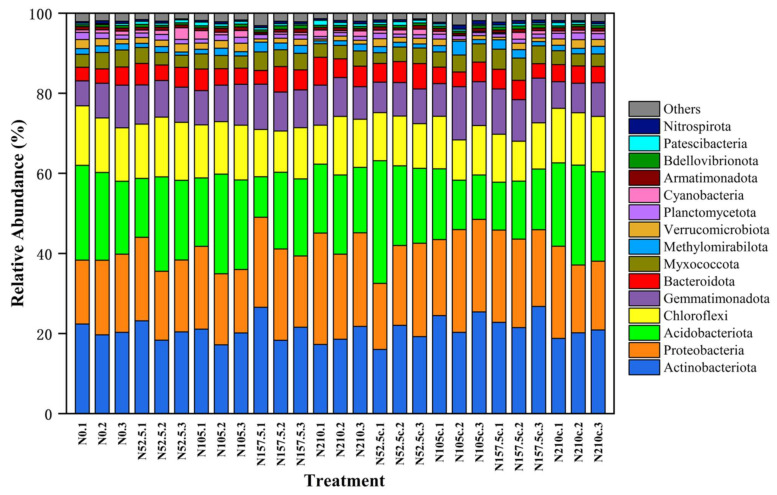
Relative abundance of top 15 soil bacterial phyla for all samples. N0, 0 kg N ha^−1^ year^−1^; N52.5, 52.5 kg N ha^−1^ year^−1^; N105, 105 kg N ha^−1^ year^−1^; N157.5, 157.5 kg N ha^−1^ year^−1^; N210, 210 kg N ha^−1^ year^−1^; N52.5c, 52.5 kg N ha^−1^ 2 year; N105c, kg N ha^−1^ 2 year; N157.5c, 157.5 kg N ha^−1^ 2 year; N210c, kg N ha^−1^ 2 year. Treatments labeled with 1, 2, and 3 represent 3 repetitions.

**Figure 3 microorganisms-10-01579-f003:**
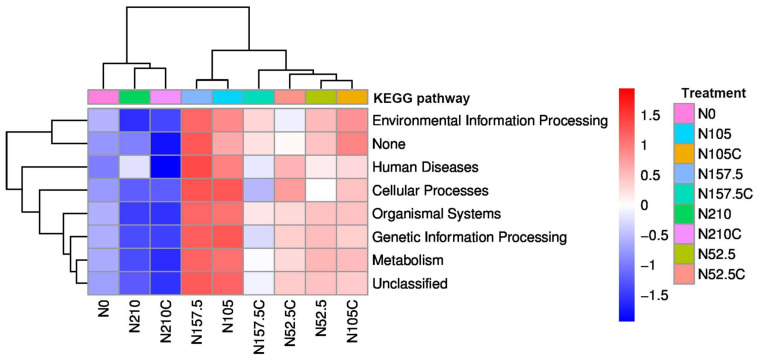
Clusters showing the relative abundance of soil bacterial communities predicted functions related to KEGG pathways the first level as affected by N fertilization. N0, 0 kg N ha^−1^ year^−1^; N52.5, 52.5 kg N ha^−1^ year^−1^; N105, 105 kg N ha^−1^ year^−1^; N157.5, 157.5 kg N ha^−1^ year^−1^; N210, 210 kg N ha^−1^ year^−1^; N52.5c, 52.5 kg N ha^−1^ 2 year; N105c, 105 kg N ha^−1^ 2 year; N157.5c, 157.5, kg N ha^−1^ 2 year; N210c, 210 kg N ha^−1^ 2 year.

**Figure 4 microorganisms-10-01579-f004:**
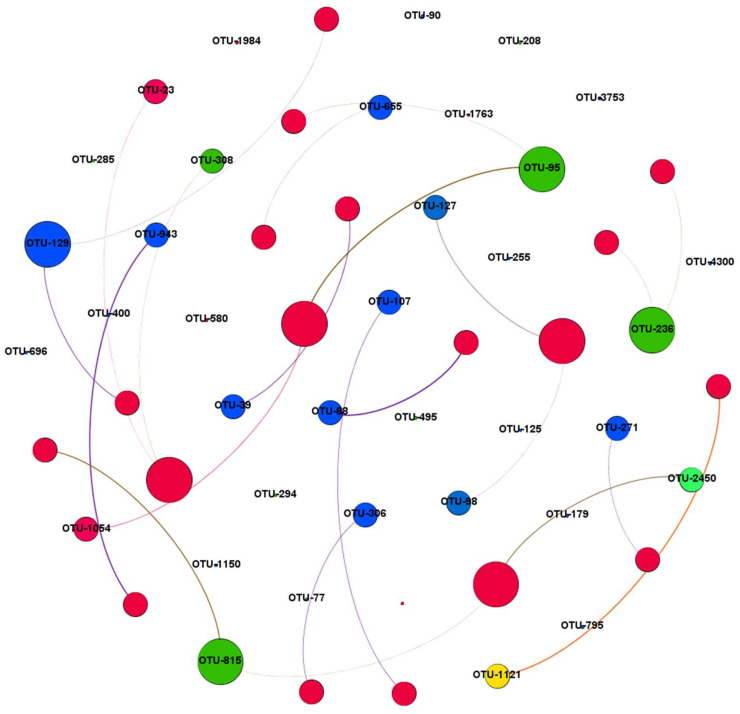
The co-occurrence network of the soil bacterial classes as affected by nitrogen (N) fertilizer treatments. Each circle represents an OTU, which are colored by modules. The size of each circle node represents its degree. The width of each connecting line represents its weight, and the colors of the edges are consistent with the colors of the nodes (circle) to which they are connected.

**Figure 5 microorganisms-10-01579-f005:**
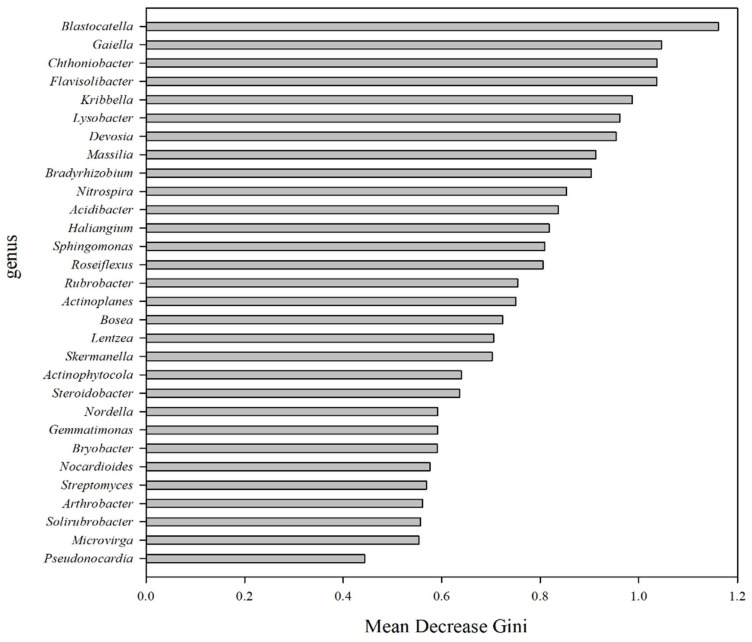
The importance of species (variables) points of the top 30 soil bacterial genera as affected by N fertilization. The abscissa is a measure of importance, and the ordinate is the name of the species sorted by importance.

**Figure 6 microorganisms-10-01579-f006:**
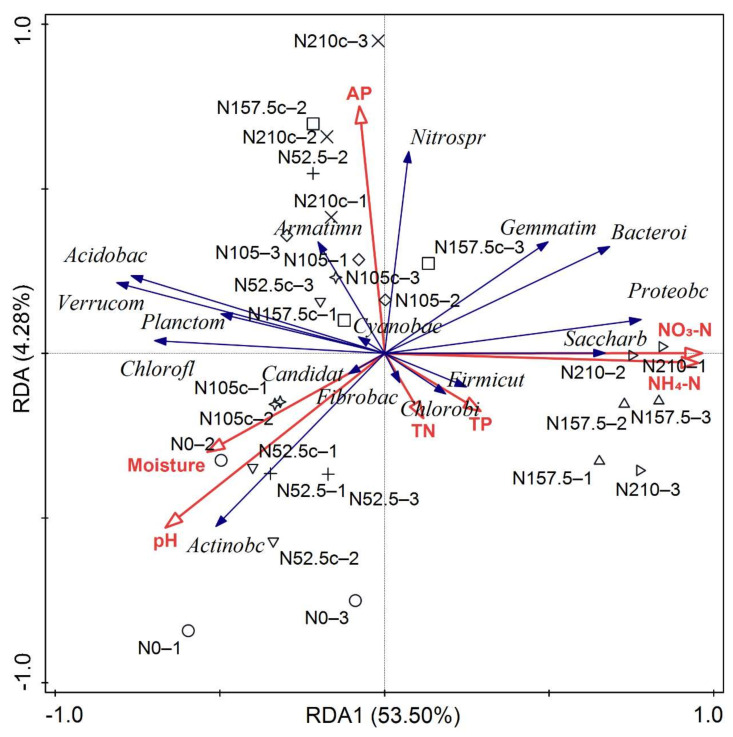
Summary of redundancy analysis (RDA) of the soil bacterial phyla. pH, soil pH; TN, soil total nitrogen; TP, soil total phosphorus; AP, soil available phosphorus. N0, 0 kg N ha^−1^ year^−1^; N52.5, 52.5 kg N ha^−1^ year^−1^; N105, 105 kg N ha^−1^ year^−1^; N157.5, 157.5 kg N ha^−1^ year^−1^; N210, 210 kg N ha^−1^ year^−1^; N52.5c, 52.5 kg N ha^−1^ 2 year; N105c, 105 kg N ha^−1^ 2 year; N157.5c, 157.5 kg N ha^−1^ 2 year; N210c, kg N ha^−1^ 2 year. Treatments labeled with −1, −2, and −3 represent 3 replicates.

**Table 1 microorganisms-10-01579-t001:** The soil bacterial communities predicted functions related to the first level of the KEGG pathway as affected by N fertilization.

Treatment	Cellular Processes	Environmental Information Processing	Genetic Information Processing	Human Diseases	Metabolism	None	Organismal Systems	Unclassified
N0	1.977 ± 0.02 a	13.644 ± 0.12 a	18.025 ± 0.043 a	0.222 ± 0.003b	42.134 ± 0.019 a	0.32 ± 0.003bc	0.765 ± 0.005 a	22.914 ± 0.046 b
N52.5	1.962 ± 0.02 a	13.683 ± 0.174 a	17.967 ± 0.139 a	0.225 ± 0.003b	42.149 ± 0.023 a	0.323 ± 0.002 abc	0.762 ± 0.003 ab	22.929 ± 0.016 b
N105	1.99 ± 0.034 a	13.555 ± 0.135 a	18.023 ± 0.11 a	0.223 ± 0.003b	42.166 ± 0.057 a	0.317 ± 0.003c	0.765 ± 0.003 a	22.96 ± 0.034 ab
N157.5	1.988 ± 0.026 a	13.634 ± 0.385 a	17.981 ± 0.133 a	0.228 ± 0.004b	42.098 ± 0.038 a	0.328 ± 0.004 abc	0.756 ± 0.002 ab	22.987 ± 0.096 ab
N210	2.031 ± 0.024 a	13.473 ± 0.182 a	17.977 ± 0.057 a	0.241 ± 0.002 a	42.099 ± 0.024 a	0.328 ± 0.003 abc	0.751 ± 0.003 b	23.1 ± 0.038 a
N52.5c	2.031 ± 0.023 a	13.603 ± 0.164 a	18.007 ± 0.111 a	0.226 ± 0.003 b	42.118 ± 0.013 a	0.322 ± 0.004 abc	0.768 ± 0.005 a	22.925 ± 0.052 b
N105c	2.011 ± 0.056 a	13.668 ± 0.101 a	17.913 ± 0.123 a	0.225 ± 0.003 b	42.136 ± 0.01 a	0.333 ± 0.005 a	0.766 ± 0.002 a	22.948 ± 0.028 ab
N157.5c	1.969 ± 0.043 a	13.735 ± 0.068 a	17.843 ± 0.026 a	0.227 ± 0.003 b	42.155 ± 0.025 a	0.331 ± 0.003 ab	0.767 ± 0.002 a	22.973 ± 0.059 ab
N210c	2.004 ± 0.01 a	13.667 ± 0.106 a	17.911 ± 0.075 a	0.222 ± 0.004 b	42.163 ± 0.025 a	0.326 ± 0.005 abc	0.766 ± 0.006 a	22.941 ± 0.033 ab
ANOVA (*p*-value)	0.726	0.991	0.919	0.013	0.626	0.11	0.073	0.315

N0, 0 kg N ha^−1^; N52.5, 52.5 kg N ha^−1^; N105, 105 kg N ha^−1^; N157.5, 157.5 kg N ha^−1^; N210, 210 kg N ha^−1^; N52.5c, 52.5 kg N ha^−1^ 2 year; N105c, 105 kg N ha^−1^ 2 year; N157.5c, 157.5 kg N ha^−1^ 2 year; N210c, 210 kg N ha^−1^ 2 year. Within a column, data (means ± SD, *n* = 3) labeled with different letters are significant at *p* < 0.05.

**Table 2 microorganisms-10-01579-t002:** The soil bacterial communities top seven predicted functions related to the second level of the KEGG pathway as affected by N fertilization.

KEGG Pathway	Treatment ^a^									
	*p*-Value	N0	N52.5	N105	N157.5	N210	N52.5c	N105c	N157.5c	N210c
**Cellular Processes**										
Cell Growth and Death	0.025	0.31 ± 0 b	0.32 ± 0 b	0.32 ± 0.01 b	0.33 ± 0 ab	0.34 ± 0 a	0.32 ± 0 b	0.32 ± 0 b	0.33 ± 0 ab	0.33 ± 0 ab
Cell Motility	0.827	1.61 ± 0.02 a	1.59 ± 0.04 a	1.62 ± 0.04 a	1.61 ± 0.03 a	1.65 ± 0.02 a	1.67 ± 0.03 a	1.64 ± 0.05 a	1.60 ± 0.04 a	1.63 ± 0.01 a
Transport and Catabolism	0.969	0.05 ± 0 a	0.05 ± 0 a	0.05 ± 0 a	0.05 ± 0 a	0.05 ± 0 a	0.05 ± 0 a	0.04 ± 0 a	0.05 ± 0 a	0.05 ± 0 a
**Environmental Information Processing**										
Membrane Transport	0.065	12.22 ± 0.02 a	12.26 ± 0.01 a	12.13 ± 0.05 ab	12.18 ± 0.08 ab	12.05 ± 0.05 b	12.21 ± 0.03 a	12.26 ± 0.05 a	12.27 ± 0.03 a	12.24 ± 0.05 a
Signal Transduction	0.409	1.25 ± 0.01 a	1.25 ± 0 a	1.24 ± 0.02 a	1.26 ± 0.02 a	1.22 ± 0.02 a	1.22 ± 0.03 a	1.23 ± 0.01 a	1.27 ± 0.02 a	1.24 ± 0.01 a
Signaling Molecules and Interaction	<0.000	0.17 ± 0 c	0.18 ± 0 b	0.18 ± 0 b	0.20 ± 0 a	0.20 ± 0 a	0.17 ± 0 c	0.18 ± 0.01 b	0.19 ± 0 b	0.18 ± 0 b
**Genetic Information Processing**										
Folding Sorting and Degradation	0.923	2.10 ± 0.01 a	2.09 ± 0.02 a	2.11 ± 0.02 a	2.09 ± 0.02 a	2.10 ± 0 a	2.11 ± 0.02 a	2.09 ± 0.03 a	2.08 ± 0.02 a	2.11 ± 0 a
Replication and Repair	0.891	5.06 ± 0.02 a	5.07 ± 0.04 a	5.04 ± 0.03 a	5.05 ± 0.05 a	5.08 ± 0.02 a	5.08 ± 0.03 a	5.04 ± 0.04 a	5.02 ± 0.01 a	5.06 ± 0.02 a
Transcription	0.718	10.86 ± 0.01 a	10.81 ± 0.08 a	10.87 ± 0.07 a	10.84 ± 0.06 a	10.79 ± 0.04 a	10.82 ± 0.07 a	10.78 ± 0.06 a	10.75 ± 0.01 a	10.74 ± 0.05 a
**Human Diseases**										
Cancers	0.177	0.06 ± 0ab	0.06 ± 0 ab	0.06 ± 0 b	0.07 ± 0 ab	0.07 ± 0 a	0.07 ± 0 ab	0.07 ± 0 ab	0.07 ± 0 ab	0.06 ± 0 b
Infectious Diseases	0.326	0.05 ± 0b	0.06 ± 0 ab	0.06 ± 0 ab	0.06 ± 0 ab	0.06 ± 0 a	0.05 ± 0 b	0.05 ± 0 b	0.06 ± 0 ab	0.06 ± 0 ab
Neurodegenerative Diseases	0.104	0.10 ± 0b	0.10 ± 0 b	0.10 ± 0 b	0.11 ± 0 ab	0.11 ± 0 a	0.10 ± 0 b	0.10 ± 0 b	0.10 ± 0 b	0.10 ± 0 b
**Metabolism**										
Amino Acid Metabolism	0.887	7.64 ± 0.01 a	7.65 ± 0.01 a	7.64 ± 0.01 a	7.64 ± 0.01 a	7.66 ± 0.02a	7.65 ± 0.01 a	7.65 ± 0.02 a	7.62 ± 0.03 a	7.63 ± 0.02 a
Biosynthesis of Other Secondary Metabolites	0.223	0.57 ± 0.01 a	0.57 ± 0 a	0.57 ± 0 a	0.56 ± 0.01 a	0.56 ± 0 a	0.57 ± 0.01 a	0.56 ± 0.01 a	0.55 ± 0 a	0.57 ± 0 a
Carbohydrate Metabolism	0.761	8.55 ± 0.08 a	8.52 ± 0.02 a	8.51 ± 0.05 a	8.48 ± 0.05 a	8.50 ± 0.06 a	8.54 ± 0 a	8.48 ± 0.05 a	8.49 ± 0.01 a	8.58 ± 0.01 a
Energy Metabolism	0.364	3.59 ± 0.01 a	3.60 ± 0.02 a	3.62 ± 0.01 a	3.58 ± 0.02 a	3.57 ± 0.01 a	3.59 ± 0.02 a	3.6 ± 0.01 a	3.59 ± 0.01 a	3.57 ± 0.01 a
Enzyme Families	0.927	3.25 ± 0.01 a	3.25 ± 0.02 a	3.27 ± 0.02 a	3.26 ± 0.01 a	3.25 ± 0.01 a	3.26 ± 0.02 a	3.26 ± 0.01 a	3.27 ± 0 a	3.27 ± 0.01 a
Glycan Biosynthesis and Metabolism	0.027	1.74 ± 0.01 a	1.74 ± 0 a	1.74 ± 0.01 a	1.69 ± 0.02 b	1.69 ± 0.02 b	1.74 ± 0.02 a	1.74 ± 0.02 a	1.75 ± 0.02 a	1.75 ± 0.01 a
Lipid Metabolism	0.390	2.44 ± 0.01 a	2.44 ± 0.02 a	2.43 ± 0.02 a	2.46 ± 0.02 a	2.44 ± 0.01 a	2.43 ± 0.02 a	2.46 ± 0.03 a	2.48 ± 0.02 a	2.42 ± 0.01 a
Metabolism of Cofactors and Vitamins	0.799	5.88 ± 0 a	5.89 ± 0.04 a	5.9 ± 0.01 a	5.84 ± 0.01 a	5.87 ± 0.01 a	5.88 ± 0.03 a	5.87 ± 0.03 a	5.85 ± 0.01 a	5.88 ± 0.04 a
Metabolism of Other Amino Acids	0.596	1.00 ± 0 a	1.01 ± 0.01 a	1.00 ± 0.01 a	1.01 ± 0.01 a	1.01 ± 0.01 a	1.01 ± 0.02 a	1.01 ± 0.01 a	1.02 ± 0 a	0.99 ± 0 a
Metabolism of Terpenoids and Polyketides	0.431	2.40 ± 0.01 a	2.41 ± 0.03 a	2.43 ± 0.03 a	2.45 ± 0.05 a	2.4 ± 0.02 a	2.41 ± 0.04 a	2.47 ± 0.03 a	2.49 ± 0.03 a	2.43 ± 0.02 a
Nucleotide Metabolism	0.360	3.34 ± 0 a	3.34 ± 0.02 a	3.36 ± 0.02 a	3.34 ± 0.03 a	3.34 ± 0.02 a	3.3 ± 0.01 a	3.3 ± 0.01 a	3.32 ± 0.02 a	3.35 ± 0.01 a
Xenobiotics Biodegradation and Metabolism	0.001	1.73 ± 0.01 b	1.74 ± 0.02 b	1.71 ± 0.01 b	1.79 ± 0.01 a	1.81 ± 0.01 a	1.74 ± 0.02 b	1.73 ± 0.02 b	1.73 ± 0.02 b	1.71 ± 0.01 b
**Organismal Systems**										
Circulatory System	<0.000	0.05 ± 0 b	0.05 ± 0 b	0.05 ± 0 b	0.05 ± 0 b	0.06 ± 0 a	0.05 ± 0 b	0.05 ± 0 b	0.05 ± 0 b	0.05 ± 0 b
Digestive System	0.455	0.03 ± 0 a	0.03 ± 0 a	0.03 ± 0 a	0.03 ± 0 a	0.03 ± 0 a	0.03 ± 0 a	0.03 ± 0 a	0.03 ± 0 a	0.03 ± 0 a
Endocrine System	0.272	0.30 ± 0 a	0.3 ± 0 a	0.30 ± 0 a	0.29 ± 0 a	0.29 ± 0 a	0.30 ± 0 a	0.29 ± 0.01 a	0.30 ± 0 a	0.29 ± 0 a
Environmental Adaptation	0.020	0.18 ± 0 a	0.18 ± 0 abc	0.18 ± 0 abc	0.18 ± 0 bc	0.18 ± 0 c	0.18 ± 0 a	0.18 ± 0 a	0.18 ± 0 ab	0.18 ± 0 a
Excretory System	0.222	0.08 ± 0 a	0.08 ± 0 a	0.08 ± 0 a	0.08 ± 0 ab	0.07 ± 0 b	0.08 ± 0 a	0.08 ± 0 ab	0.08 ± 0 ab	0.08 ± 0 ab
Nervous System	0.854	0.13 ± 0 a	0.13 ± 0 a	0.13 ± 0 a	0.13 ± 0.01 a	0.13 ± 0 a	0.13 ± 0 a	0.14 ± 0.01 a	0.14 ± 0.01 a	0.13 ± 0.01 a
**Unclassified**										
Cellular Processes and Signaling	0.003	5.53 ± 0.01 b	5.53 ± 0.01 b	5.58 ± 0.01 b	5.61 ± 0.01 b	5.7 ± 0.01 a	5.54 ± 0.03 b	5.57 ± 0.03 b	5.58 ± 0.02 b	5.57 ± 0.05 b
Genetic Information Processing	<0.000	4.15 ± 0.01 d	4.21 ± 0.01 a	4.22 ± 0.01 a	4.17 ± 0 c	4.18 ± 0.01 bc	4.18 ± 0.01 bc	4.17 ± 0.01 bc	4.18 ± 0 bc	4.20 ± 0 ab
Metabolism	0.409	4.27 ± 0.00 a	4.29 ± 0.01 a	4.27 ± 0.01 a	4.28 ± 0 a	4.29 ± 0.01 a	4.28 ± 0 a	4.27 ± 0.01 a	4.27 ± 0.01 a	4.28 ± 0.01 a
Poorly Characterized	0.363	8.95 ± 0.01 a	8.90 ± 0.02 a	8.89 ± 0.01 a	8.93 ± 0.02 a	8.92 ± 0.02 a	8.92 ± 0.02 a	8.93 ± 0.02 a	8.94 ± 0.01 a	8.90 ± 0.03 a

^a^ N0, 0 kg N ha^−1^ year^−1^; N52.5, 52.5 kg N ha^−1^ year^−1^; N105, 105 kg N ha^−1^ year^−1^; N157.5, 157.5 kg N ha^−1^ year^−1^; N210, 210 kg N ha^−1^ year^−1^; N52.5c, 52.5 kg N ha^−1^ 2 year; N105c, 105 kg N ha^−1^ 2 year; N157.5c, 157.5 kg N ha^−1^ 2 year; N210c, 210 kg N ha^−1^ 2 year. Data (means ± SD, *n* = 3) labeled with different letters are significant at *p* < 0.05.

**Table 3 microorganisms-10-01579-t003:** Soil physiochemical parameters as affected by N fertilization.

Treatment	pH	TN (g kg^−1^)	NH_4_-N (mg kg^−1^)	NO_3_-N (mg kg^−1^)	TP (g kg^−1^)	AP (mg kg^−1^)	SM (%)
N0	8.09 ± 0.04 ^a^	0.92 ± 0.05 ^a^	11.61 ± 0.20 ^b^	15.74 ± 0.12 ^f^	0.60 ± 0.03 ^a^	21.52 ± 1.44 ^c^	11.63 ± 0.91 ^a^
N52.5	7.94 ± 0.03 ^bc^	0.97 ± 0.13 ^a^	12.05 ± 0.51 ^b^	19.77 ± 0.55 ^de^	0.70 ± 0.09 ^a^	23.74 ± 0.55 ^b^	11.57 ± 0.46 ^a^
N105	7.87 ± 0.03 ^cd^	1.09 ± 0.03 ^a^	12.68 ± 0.32 ^b^	26.08 ± 0.63 ^b^	0.77 ± 0.06 ^a^	26.66 ± 0.21 ^a^	11.45 ± 0.27 ^a^
N157.5	7.81 ± 0.01 ^d^	1.05 ± 0.04 ^a^	18.09 ± 0.24 ^a^	46.58 ± 0.68 ^a^	0.77 ± 0.08 ^a^	23.55 ± 0.69 ^bc^	10.75 ± 0.50 ^a^
N210	7.83 ± 0.05 ^d^	0.99 ± 0.01 ^a^	18.69 ± 0.89 ^a^	47.35 ± 0.42 ^a^	0.76 ± 0.07 ^a^	23.94 ± 0.83 ^b^	10.55 ± 0.22 ^a^
N52.5c	7.98 ± 0.01 ^b^	1.00 ± 0.02 ^a^	11.82 ± 0.12 ^b^	18.57 ± 0.69 ^e^	0.75 ± 0.02 ^a^	23.93 ± 0.72 ^b^	11.86 ± 0.67 ^a^
N105c	7.93 ± 0.01 ^bc^	1.02 ± 0.05 ^a^	11.37 ± 0.31 ^b^	20.39 ± 0.66 ^d^	0.71 ± 0.08 ^a^	23.98 ± 0.56 ^b^	11.64 ± 0.64 ^a^
N157.5c	7.87 ± 0.03 ^cd^	0.98 ± 0.02 ^a^	12.74 ± 0.25 ^b^	22.72 ± 0.65 ^c^	0.72 ± 0.16 ^a^	25.02 ± 0.31 ^ab^	10.82 ± 0.52 ^a^
N210c	7.82 ± 0.04 ^d^	1.00 ± 0.02 ^a^	12.81 ± 0.55 ^b^	23.26 ± 0.61 ^c^	0.61 ± 0.11 ^a^	26.28 ± 0.25 ^a^	10.73 ± 0.25 ^a^
ANOVA *p*-value	<0.001	0.557	<0.001	<0.001	0.799	0.003	0.564

N0, 0 kg N ha^−1^ year^−1^; N52.5, 52.5 kg N ha^−1^ year^−1^; N105, 105 kg N ha^−1^ year^−1^; N157.5, 157.5 kg N ha^−1^ year^−1^; N210, 210 kg N ha^−1^ year^−1^; N52.5c, 52.5 kg N ha^−1^ 2 year; N105c, 105 kg N ha^−1^ 2 year; N157.5c, 157.5 kg N ha^−1^ 2 year; N210c, 210 kg N ha^−1^ 2 year. TN, total N; TP, total phosphorus; AP, available phosphorus; SM, soil moisture. Data (means ± SD, *n* = 3) labeled with different letters are significant at *p* < 0.05.

**Table 4 microorganisms-10-01579-t004:** Pearson’s correlations between the bacterial phylum and soil properties.

Taxonomy	NH_4_-N	NO_3_-N	TN	TP	AP	pH	Soil Moisture
*Proteobacteria*	0.755 **	0.752 **	0.013	0.189	0.003	−0.568 **	−0.419 *
*Actinobacteria*	−0.392 *	−0.444 *	0.056	−0.214	−0.305	0.568 **	0.570 **
*Acidobacteria*	−0.709 **	−0.724 **	−0.151	−0.289	0.271	0.384 *	0.402 *
*Chloroflexi*	−0.715 **	−0.732 **	−0.239	−0.246	−0.116	0.464 *	0.259
*Gemmatimonadetes*	0.406 *	0.519 **	0.358	−0.033	0.182	−0.643 **	−0.426 *
*Bacteroidetes*	0.652 **	0.646 **	0.059	0.057	0.201	−0.637 **	−0.542 **
*Nitrospirae*	0.052	0.124	0.368	0.077	0.723 **	−0.558 **	−0.118
*Verrucomicrobia*	−0.862 **	−0.836 **	−0.059	−0.078	0.187	0.388 *	0.296
*Planctomycetes*	−0.395 *	−0.454 *	−0.198	−0.301	0.169	0.295	0.311
*Cyanobacteria*	−0.120	−0.096	−0.003	0.036	0.113	0.167	−0.182
*Armatimonadetes*	−0.220	−0.190	−0.198	−0.047	0.256	0.048	0.002
*Saccharibacteria*	0.658 **	0.712 **	0.378	0.245	0.182	−0.488 **	−0.314

TN, total N; TP, total phosphorus; AP, available phosphorus. Correlation coefficients labeled with * and ** are significant at *p* < 0.05 and *p* < 0.01, respectively.

## Data Availability

The raw data are available at the NCBI SRA archive (BioProject ID: PRJNA863891).

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
