# Peer review of "Bacterial Diversity and Potential Functions in Response to Long-Term Nitrogen Fertilizer on the Semiarid Loess Plateau"

_microorganisms, 2022, doi:10.3390/microorganisms10081579_

Round 1
Reviewer 1 Report
1. The relevance, importance and innovative aspects of the paper should be more evident.
2. Please present a strong case for how this paper is a major advance. This needs to be done in the manuscript itself, not just in the response to review comments.
3. Please be sure that the abstract and the conclusions section not only summarize the key findings of the work but also explain the specific ways in which this work fundamentally advances the field relative to prior literature.
4. The introduction needs to be clear on that point and can/must be extended quit a bit.
5. The aim of the research is not clearly defined. Add correct hypothesis and correlate with objective clearly.
6. The methodology should be described in more detail. In the ,Materials and methods’ section the authors did not indicate the sources of literature different methods used.
7. Why the analyzes concern only one period in one year?
8. Table no. 2 is not readable. Too much information causes its illegibility.
9. Discussion chapter - This seems to be the core of the manuscript. You should make it much more clear.
10. Please check whether you need that many references. Make sure the references are added correctly according to the journal's instructions.
11. The language correctness should be verified by a native speaker.
Reviewer 2 Report
The authors have done a great and interesting study. The data is obviously applicable in agriculture.
It seems to me that the article can be published with minor edits. In particular, it is necessary to clarify what the footnotes in Table 4 mean. Table 2 is very hard to read.
The authors obtained information at the level of different taxa up to the genus. This is typical for the metagenomic approach. However, there is a question: did you try to isolate cultures and deal with the species diversity of bacteria in soils? This information could be valuable in assessing the functional role of the community.
Reviewer 3 Report
The article covers relevant topics and is worthy of publication in this issue. Some inaccuracies were noted.
ABSTRACT. The abstract reflects well the content of the entire article, only Latin names should be in italic.
INTRODUCTION. The introduction and background are sufficient to demonstrate the study. But in paragraph - lines 65–70, and also elsewhere, leaching of N and other elements should be considered.
MATERIALS AND METHODS. All methods are well described. As should be understood, crop rotation was not applied in the long-term experiment.
Line 104: The phenological development stages of plants according to BBCH-scale must be indicated – ,,flowering” alone is not enough.
2.3. Data analysis: More information on alpha and beta diversity may be added.
RESULTS AND DISCUSSION. Results and Discussion parts are also written well, but there are no pictures and tables referred to in the text: Figure S2a, Figure S2b; Figure S3; Table S1, Table S2. Maybe they are provided in supplements?
Line 249 latin names in italic, there is and confusion with Figure 4, in the text is Figure S4 (line 250), but presented Figure 4 (line 252).
For Figure 5 should have improved resolution.
Line 177 ,, treatments was decreased by 40–57% and 92–115% respectively” – what does 115% mean, there can't be a percentage more than 100?
Line 185 – the same: was increased by 8.05–143.7% ??? Probably the authors meant a 0,8–1.4 times increase, but not %.
Throughout the text in many places, the spelling of Latin names needs to be corrected, they should be in Italian font.
The authors did not provide link to the raw NGS data in NCBI database.
Round 2
Reviewer 1 Report
Accept in present form.